# Specific Autoantibodies and Microvascular Damage Progression Assessed by Nailfold Videocapillaroscopy in Systemic Sclerosis: Are There Peculiar Associations? An Update

**DOI:** 10.3390/antib12010003

**Published:** 2023-01-04

**Authors:** Elvis Hysa, Rosanna Campitiello, Silvia Sammorì, Emanuele Gotelli, Andrea Cere, Giampaola Pesce, Carmen Pizzorni, Sabrina Paolino, Alberto Sulli, Vanessa Smith, Maurizio Cutolo

**Affiliations:** 1Laboratory of Experimental Rheumatology and Academic Division of Clinical Rheumatology, Department of Internal Medicine, University of Genoa, San Martino Polyclinic, 16132 Genoa, Italy; 2Autoimmunity Laboratory, Department of Internal Medicine, University of Genoa, 16132 Genoa, Italy; 3Autoimmunity Diagnostic Laboratory, IRCCS San Martino Polyclinic, 16132 Genoa, Italy; 4Department of Internal Medicine, Ghent University, 9000 Ghent, Belgium; 5Department of Rheumatology, Ghent University Hospital, 9000 Ghent, Belgium; 6Unit for Molecular Immunology and Inflammation, VIB Inflammation Research Centre (IRC), 9052 Ghent, Belgium

**Keywords:** systemic sclerosis, autoantibodies, microangiopathy

## Abstract

Background: Specific autoantibodies and nailfold videocapillaroscopy (NVC) findings are serum and morphological diagnostic hallmarks of systemic sclerosis (SSc) as well as useful biomarkers which stratify the microvascular progression and prognosis of patients. Methods: The aim of our narrative review is to provide an update and overview of the link between SSc-related autoantibodies, used in clinical practice, and microvascular damage, evaluated by NVC, by exploring the interaction between these players in published studies. A narrative review was conducted by searching relevant keywords related to this field in Pubmed, Medline and EULAR/ACR conference abstracts with a focus on the findings published in the last 5 years. Results: Our search yielded 13 clinical studies and 10 pre-clinical studies. Most of the clinical studies (8/13, 61.5%) reported a significant association between SSc-related autoantibodies and NVC patterns: more specifically anti-centromere autoantibodies (ACA) were associated more often with an “Early” NVC pattern, whereas anti-topoisomerase autoantibodies (ATA) more frequently showed an “Active” or “Late” NVC pattern. Five studies, instead, did not find a significant association between specific autoantibodies and NVC findings. Among the pre-clinical studies, SSc-related autoantibodies showed different mechanisms of damage towards both endothelial cells, fibroblasts and smooth muscle vascular cells. Conclusions: The clinical and laboratory evidence on SSc-related autoantibodies and microvascular damage shows that these players are interconnected. Further clinical and demographic factors (e.g., age, sex, disease duration, treatment and comorbidities) might play an additional role in the SSc-related microvascular injury whose progression appears to be complex and multifactorial.

## 1. Introduction

Systemic sclerosis (SSc) is a complex autoimmune connective tissue disease, characterized by microvascular damage, aberrant immune response, production of autoantibodies and progressive fibrosis of skin and internal organs [1].

SSc patients might display heterogeneous clinical phenotypes beginning from the extension of skin thickening to the autoantibody profile, disease progression and organ involvement: most commonly gastrointestinal tract, lung, kidneys and heart [2,3,4,5].

The most frequent symptom of sclerodermic microangiopathy is Raynaud’s phenomenon (RP) which is present in up to 96% of patients, being one of the earliest manifestations of the disease [6].

To date, the identification of patients at an early stage may represent a challenge for physicians, nevertheless, it guides to a tailored approach in terms of follow-up and treatment.

In this respect, nailfold videocapillaroscopy (NVC) and SSc-specific autoantibodies represent useful tools for the early diagnosis of SSc patients [7,8,9]. NVC allows the identification of peripheral microvascular damage in SSc patients and the “scleroderma pattern” on NVC has been incorporated into the 2013 American College of Rheumatology (ACR)/European Alliance of Associations for Rheumatology (EULAR) classification criteria of SSc [10]. The microangiopathy in SSc is described with a typical progression going through three patterns of microvascular damage: the “Early”, characterized by the presence of giant capillaries with a preserved capillary density, the “Active”, featured by the association of giant capillaries combined with a lower capillary density, and the “Late” which displays a further lowered capillary density combined with abnormally shaped capillaries, expression of neoangiogenesis [8] (Figure 1). Indeed, the combination of a “scleroderma pattern” and SSc-specific autoantibodies displays the highest performance in discerning, in a Raynaud’s phenomenon population, who will and who will not develop SSc [11]. Three autoantibodies, highly specific for SSc, are included in the classification criteria: anti-centromere (ACA), anti-topoisomerase (ATA) and anti-RNA-polymerase III (ARA).

With the advent of more sensitive multiplexed immunoassays, other autoantibodies have been described with a varying sensitivity and specificity in SSc cohorts depending on the ethnicity, geographic region, immunogenetic markers and autoantigen/immunoassay used [12,13].

For ATA and ACA there is a good agreement between the different immune-assays whereas the inclusion of ARA is based on more limited evidence due to a different agreement between the immune-assays [12]. Conversely, the lower prevalence of the other SSc-associated autoantibodies hampers, at the moment, the creation of solid data to support their inclusion in the classification criteria generating the need for a harmonization of immune-assays in terms of interpretation.

Despite these technical limitations, disease-specific autoantibodies are pivotal biomarkers in SSc, due to their ability to stratify patients with different severity and prognosis [12,14,15].

For instance, anti-Th/To define a phenotype of patients often characterized by limited cutaneous disease and a variable rate of cardio-pulmonary involvement (i.e., interstitial lung disease (ILD) or pericarditis) [12]. Many other autoantibodies could be considered markers of overlap syndrome with idiopathic inflammatory myopathies (IIMs). Anti-Ku are found in 2–7% of SSc patients presenting also clinical features of polymyositis (PM) or PM-Scl autoantibodies which define SSc patients with high frequency of ILD, calcinosis, dermatomyositis skin alterations and severe muscle involvement. Finally, novel SSc specific autoantibodies have been described in about 10% of the former called “seronegative” SSc patients: anti-elF2B, anti-RuvBL1/2 complex, anti-U11/U12 RNP and anti-BICD2 depict specific SSc subtypes with severe organ complications [12].

Besides their usefulness in stratifying different phenotypes of patients which allows a personalized medicine, there is growing, despite conflicting, evidence of their pathogenicity in inducing the microvascular damage, detectable on NVC, and generalized organ damage [16,17].

In this review, we synthetized qualitatively the results of papers reporting the association of SSc-specific antibodies and structural microvascular abnormalities evaluated with NVC, discussing the evidence in light of the most recent literature.

## 2. Methods

A literature search was performed on PubMed and Medline using these terms: systemic sclerosis, microscopic angioscopy, autoantibodies, autoimmunity and microvessels. The search was applied to all studies published in English. The archive of the conference abstracts of The European Alliance of Associations for Rheumatology (EULAR) and American College of Rheumatology (ACR) of the last three years (2019–2022) was reviewed as well for pertinent publications. Additionally, the reference list of the included papers was scanned for additional publications meeting this study’s aim. When papers reported data partially presented in previous articles, we referred to the most recent published data.

R.C. and S.S. performed, in double, the literature search by including the clinical studies, which addressed NVC and autoantibody findings in SSc patients, and translational manuscripts on the damage mediated by SSc-specific autoantibodies on the endothelium layer cellular components (i.e., endothelial cells, fibroblasts, smooth muscle cells). This process was supervised by E.H. and M.C. who solved the conflicts when consensus was not reached by the researchers. Finally, the data were extracted in form of standardized tables by E.H. and R.C.

The results of the search are reported as a narrative review with a focus on the most recent evidence published in the last 5 years.

## 3. Results

### 3.1. Evidence of the Association between SSc-Specific Autoantibodies and Microvascular Damage Detected by NVC in Clinical Studies

In total, 13 papers reported the association between SSc-related autoantibodies and NVC findings (Table 1).

In eight reports, significant differences in NVC patterns were detected according to the autoantibody profile: the most explored autoantibodies were ACA and ATA. In an early report by Chandran et al., ATA+ patients showed more frequent capillary loss compared with ACA+ who displayed more frequent capillary dilations or giant capillaries [18]. Similar results were reported by Cutolo et al., in a cohort of 241 patients, concluding that ATA+ patients showed more frequent “Active” and “Late” NVC patterns than “Early”, suggesting a faster progression of the microangiopathy in this subgroup of patients [19]. Indeed, the authors concluded that the presence of ATA appeared to be related to an earlier expression of the “Active” and “Late” NVC patterns whereas the presence of ACA seems to be related to a delayed expression of the “Late” NVC pattern [19]. Analogue conclusions in terms of frequencies of “Early”, “Active” and “Late” patterns were reported by Sulli et al., and Pizzorni et al. [20,21]. These findings were later supported by a detailed analysis of 2754 SSc patients from the EULAR scleroderma trials (EUSTAR) database where Ingegnoli et al. reported that the “Late” pattern was present in 47% of ATA+ patients vs. 28% of ACA+ patients (*p* < 0.05) whereas the “Early” and “Active” patterns were more frequent in ACA+ than in ATA+ patients (44% vs. 28%, *p* < 0.05). Significant associations were found between ATA positivity and “late” SSc pattern, detected in 47% of patients, and between ACA positivity with higher prevalence in the “Early” and “Active” SSc pattern vs. the “Late” Pattern (45% vs. 43% vs. 27%, *p* = 0.03).

In more recent papers, NVC findings of SSc patients with other specific autoantibody subtypes were explored. Tieu et al., evaluated the capillary total damage index and the capillary loss in 152 SSc patients and reported a significantly higher nailfold capillary total damage index in ARA+ patients compared with ACA+ and RNP+ patients (*p* < 0.01 for both comparisons) [22]. Interestingly, both ARA+ and ATA+ patients had a greater capillary drop out than ACA+ patients despite having significantly shorter disease duration.

This result on ARA+ patients conflicts with another recent cohort where ARA positivity was associated with a higher mean capillary density with an absence of “Active” and “Late” NVC patterns [23]. As a matter of fact, it is necessary to emphasize that in this latter cohort the small sample size of patients (*n* = 19) might have impaired the subgroup analysis of patients according to their autoantibody profile.

Lastly, in the largest study investigating SSc-specific autoantibodies and NVC findings as predictors of development of definite SSc in a RP population “at risk”, all SSc-specific antibodies, specifically ACA, anti-Th/To and ARA predicted capillary loss (HR 2.62) [11].

On the other hand, there is also evidence reporting non-significant associations between autoantibodies and NVC findings, either defined as “normal” patterns or “SSc-patterns” or as “Early”, “Active” and “Late” patterns. In four out of five papers (80%) reporting an absence of significant associations between autoAb and NVC, the analyzed autoantibodies were ACA and ATA: two papers failed to report an association between these antibodies with “Early”, “Active” and “Late” patterns [24,25] or with NVC findings dichotomously defined as “Normal” vs. “SSc-pattern” [26,27].

A recent article failed also in detecting significant associations between SSc-specific antibodies including an enlarged panel of 11 antigens with NVC specific findings classified as “Early”, “Active” and late patterns [28].

In conclusion, most of the papers (8/13, 61.5%) indicate an association between SSc-specific antibodies and microvascular damage but, as reported in a recent systematic literature review investigating the association between sex and antibodies with microvascular damage in SSc [29], a quantitative synthesis through meta-analysis, besides being beyond the aims of our narrative review, would also be theoretically difficult because of a wide heterogeneity in the definitions of the microvascular damage assessed by NVC, different antibodies assessed, methods used for their detection and, to a lesser extent, diversity in the classification criteria used for defining SSc patients.

### 3.2. Evidence of Microvascular Damage Mediated by SSc-Specific Autoantibodies in Pre-Clinical Studies and Translational Research

Overall, 10 papers reported the association between SSc-related autoantibodies and microvascular damage suggesting their potential involvement in SSc pathophysiology (Table 2).

The most explored autoantibodies were ACA and ATA with compelling data indicating their interaction with endothelial cells and fibroblasts in in vitro studies.

The correlation between ACA and vascular damage has also been explored in two in vitro early studies. Extracellular centromeric protein B (CENP-B) autoantigens were found to specifically bind the surface of human pulmonary artery smooth muscle cells (PASMCs) stimulating their migration and their production of local IL-6 and IL-8, which display an important physiological role in the first stages of local endothelial tissue repair [31,32]. Since CENP-B behaves as a potent stimulation of PASMCs activating the endothelial growth factor receptor (EGFR), anti–CENP-B antibodies were found to prevent CENP-B from transactivating EGFR and exerting its cytokine-like activities toward vascular smooth muscle cells. These investigated mechanisms shed light on the possible role of CENP-B/ACA immune complexes in the initial aberrant tissue repair processes observed in SSc patients [31].

Contrarily, topoisomerase I (Topo I) autoantigen was shown to display a higher affinity for fibroblasts compared with endothelial and smooth muscle cells. Indeed, Topo I binds to the cell surface of fibroblast and recruits ATA as detected by indirect immunofluorescence. Subsequently, ATA promote monocytes adhesion to fibroblasts followed by the up-regulation of monocytes’ markers (CD14 and CD11b) [33]. Another wound healing assay of fibroblasts suggested a dose-dependent role of Topo I in stimulating fibroblast migration. This process is mediated via a G-protein-coupled receptor which activates MAPK intracellular signaling pathways [34]. Moreover, Topo I binds to heparan-sulfate (HS) proteoglycans on normal human dermal fibroblasts (HDFs) and ATA deriving from sera of SSc patients amplify Topo I binding to HS chains enhancing the fibrotic process [35].

More recently, the aforementioned “dichotomy” of ACA interfering with the endothelium and ATA impairing fibroblasts’ activity was overcome by data suggesting that both ACA and ATA can directly induce pro-fibrotic activation of fibroblasts through a hyper-expression of pro-fibrotic genes [36]. This has been documented by increased detected mRNA concentrations of ACTA2, COL1A1 and TAGLN and upregulated protein synthesis of α-SMA, Col-1 and SM22, all of them being connective tissue proteins [36].

Additionally, SSc-specific autoantibodies (ATA, ACA, ARA and Th/To) embedded in immune complexes (ICs) displayed pro-inflammatory and/or pro-fibrotic effects on endothelial cells in a cohort of 12 SSc patients with slightly different properties between them [37] (for more details see Table 2). Indeed, these ICs were shown to induce an increase in Intercellular Adhesion Molecule 1 (ICAM-1) expression, interleukin-8 (IL-8), Toll-like receptors (TLRs 2–4 and 9), endothelin-1 and TGF-β1 from endothelial cells [37]. Interestingly, when SSc-IC interacted with dermal fibroblasts, an increased expression of TGF-beta1 and alfa-SMA expression was observed and all SSc-Ab-ICs, but ACA-ICs augmented IL-6 levels [37].

Fewer data are available for ARA pathogenicity on endothelial cells. Indeed, we have indirect proof of a potential ARA-mediated endothelial damage highlighted by Fonceca et al., who reported a positive correlation between ARA positivity and endothelin receptors (EDNR) polymorphisms [38]. This group observed increased frequencies of EDNRA polymorphic alleles in ARA+ SSc patients compared with ARA- patients and healthy controls. This link between endothelin axis polymorphisms and ARA positivity might partly explain why ARA+ patients display a higher risk for developing scleroderma renal crisis and a rapid diffuse skin involvement.

With the extent of the newer SSc-specific antibodies in clinical practice, new pathogenetic insights have emerged. Recently, Svegliati et al. analyzed the action of antibodies against platelet-derived growth factor alpha (PDGFRα) as a key component in the activation of human pulmonary artery smooth muscle cells (HPASMC) inducing their increase in the production of radical oxygen species (ROS) and an upregulated expression of NADPH Oxidase 4 (NOX4) and mammalian target of rapamycin complex 1 (mTORC1) [39]. Ultimately, HPASMC acquired a synthetic state featured by a higher growth rate, migratory activity, gene expression of type I collagen α1 chain leading to intimal hyperplasia.

To conclude, we provided here the evidence of pathogenicity of SSc-related autoantibodies on the endothelial/fibroblastic interface to elaborate on a cellular and molecular level what we can see morphologically on the NVC findings. The targets of these antibodies are complex and, besides interfering with physiological processes carried by the released autoantigens, they are endowed with functional properties since they activate signaling pathways activating different membrane receptors (Table 2 and Figure 2).

## 4. Discussion

This review explored the interaction between SSc-specific autoantibodies, commonly searched in current clinical practice, and microvascular damage which represents one of the key pathogenetic steps in SSc assessable through a well-studied progression by NVC which is specific compared to other autoimmune connective tissue diseases [41,42,43].

It is a consolidated concept that NVC findings and SSc-related autoantibodies are independent prognostic markers for patients but their inter-connection is not completely clear because of the complex pathophysiology of SSc which involves, besides B-cells autoimmunity, the aberrant activation of T cells, pro-fibrotic M2 polarized macrophages and dendritic cells which orchestrate the damage of the endothelium and the hyper-activation of fibroblasts [44,45].

Among the clinical studies correlating SSc-specific autoantibodies and NVC findings we reported five papers highlighting the absence of significant association between the two variables. In this respect, it can be perceived by intuition that there might subsets of patients where microvascular damage might be driven mainly by B-independent pathways.

Besides the clinical impression that ACA+ patients show a slower microvascular and clinical progression compared with ATA+ and ARA+ individuals, there is the availability of strong data deriving from large European databases, such as EUSTAR, which support the concept that autoantibody profile, cutaneous involvement and NVC findings outline different pathways for each patient [30].

Furthermore, the interaction previously described between antibodies and different cellular targets such as endothelial cells and fibroblasts seems to suggest that these autoantibodies may not be only an epiphenomenon of tissue damage but also as pathogenetic agents directly involved in endothelial cumulative damage and consequential fibrosis [46,47]. To make some examples, while ACA positivity is mainly associated with consistent vascular injury, a limited cutaneous involvement, pulmonary arterial hypertension without lung fibrosis, subcutaneous calcinosis and long-standing RP [48,49], ATA+ patients display more frequently diffuse cutaneous involvement, interstitial lung disease (ILD), gastrointestinal involvement, heart fibrosis, digital ulcers and hand disabilities due to metacarpophalangeal (MCF) and proximal interphalangeal (IFP) joints flexion contractures [50,51]. Moreover, hypotheses related to the link between autoantibodies and microvascular damage in SSc derive also from the insights inherent to functional antibodies described in SSc patients whose most relevant identified targets appear to be endothelin 1 type A receptor (ETAR), angiotensin II type 1 receptor (AT1R), muscarinic receptor 3 (M3R), PDGFR, chemokine receptors CXCR3 and CXCR4, estrogen receptor α and cluster of differentiation 22 (CD22) [52].

They are called “functional” autoantibodies because, through the binding of these antigens (i.e., receptors), these antibodies can directly activate or inhibit a molecular pathway which can be replicated in an experimental setting.

Although some of these antibodies are not specific for SSc being detectable also in patients with other vascular diseases (i.e., idiopathic pulmonary arterial hypertension), Riemekasten et al. reported that higher levels of anti-AT1R and anti-ETAR autoantibodies were associated with severe SSc vascular manifestations, including digital ulcers, PAH and renal crisis [53].

Other potential indirect proofs of a direct endothelial damage mediated by these functional antibodies derives from the current therapeutical approaches of severe vascular damage in SSc such as digital ulcers or scleroderma renal crisis which rely on endothelin receptor blockers (i.e., bosentan) or angiotensin converting enzyme (ACE) inhibitors such as captopril [54,55,56]. Indeed, many questions might be raised as to whether these agents might block the agonistic effects of anti-AT1R and anti-ETAR, even though direct evidence is lacking [57,58].

The molecular mechanisms, through which these antibodies act, involve intracellular signaling mediated by G-proteins which terminate in the increased transcription and synthesis of pro-inflammatory and pro-fibrotic mediators by endothelial cells and immune system cells (such as neutrophils and T cells) which impair the regulation of angiogenesis and enhance fibrosis from fibroblasts [59].

However, further research is needed to draw definite conclusion with respect to any pathogenic role in SSc since there are limitations related to the lack of disease specificity for anti-AT1R and anti-ETAR, the use of pooled total serum immunoglobulin G (IgG) rather than affinity purified anti-AT1R and anti-ETAR in the various in vitro and in vivo experiments, and cross-reactivity between AT1R and ETAR [52].

Despite the previous reported evidence about the association between SSc-autoantibodies in clinical and translational research, the literature findings going on the opposite direction suggest how scleroderma-associated microangiopathy might be driven by other complex factors beyond autoantibodies which can be confounders in clinical studies: age, sex, disease duration, comorbidities and treatment.

Although sex does not seem to influence the degree of microangiopathy in SSc according to a recent systematic literature review [29], the other variables need to be weighted in clinical studies to achieve a better understanding of which factors most influence microvascular progression in SSc.

The study of these correlations would not only have a theoretical value considering that microvascular status strongly correlated with overall organ damage in SSc [60].

In this respect, large international studies, such as the ones deriving from the EUSTAR database are going in the direction of a more precise stratification of patients and with the accumulation of more data, the question inherent the relationship of autoantibodies and microvascular damage might have, in the future, an answer weighted by other clinical and demographic factors.

## 5. Conclusions

Autoantibodies and NVC findings are serum and morphological biomarkers which stratify the prognosis of SSc patients. Particularly, a faster microvascular progression and certain autoantibody profiles (i.e., ATA+ or ARA+ patients) characterize patients with worse clinical outcomes.

Despite the conflicting results of clinical studies, translational research supports the evidence of a direct microvascular damage mediated by antibodies such as ACA and ATA.

However, the clinical findings suggest that even other important clinical and demographic factors might influence the progression of microvascular injury in SSc worthy of further detailed analysis.

## Figures and Tables

**Figure 1 antibodies-12-00003-f001:**
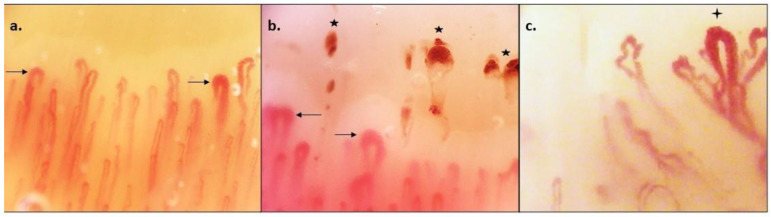
Phases of SSc-related microangiopathy detectable through nailfold videocapillaroscopy (NVC). (**a**). “Early” pattern: few giant capillaries (arrows) with a preserved capillary density, co-existence of normal and dilated capillaries. (**b**). “Active” pattern: association of giant capillaries (arrows)combined with a lower capillary density and presence of microhaemorrhages (stars). (**c**). “Late” pattern: further lowered capillary density and abnormal shapes (cross), expression of neoangiogenesis. (The images are a courtesy of Prof. Cutolo and derive from the NVC central database of The Rheumatology Unit of Genoa University).

**Figure 2 antibodies-12-00003-f002:**
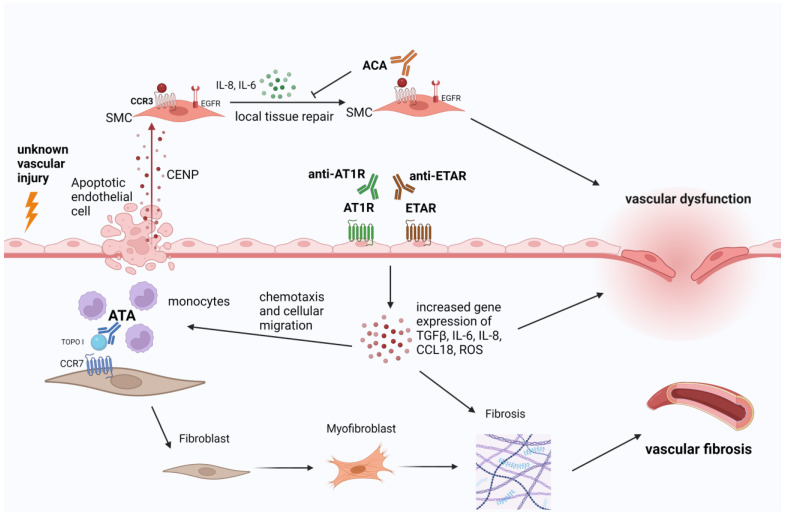
Summarized mechanisms of microvascular injury mediated by autoantibodies in systemic sclerosis deriving from pre-clinical studies. CENP (centromeric protein B), SMC (smooth muscle cell), CCR- (C-C Motif Chemokine Receptor), EGFR (endothelial growth factor receptor), IL- (interleukin), ACA (anticentromere antibodies), AT1R (angiotensin 1 receptor), ETAR (endothelin 1 type A receptor), ATA (anti-topoisomerase 1 autoantibodies), TOPO 1 (topoisomerase 1), ROS (reactive oxygen species), CCL- (C-C-motif ligand).

**Table 1 antibodies-12-00003-t001:** Papers investigating the association between SSc-related autoantibodies and NVC patterns.

Author, Year [Reference]	Population	Sample size (*n*) Sex (F: %) Age (M ± SD) Duration of Disease or Symptoms in Years, Average (Range)	Autoantibodies	Immunoassays	Results
Chandran, 1995 [18]	SSc patients, defined with diagnosis reported in the clinical files	*n* = 52 F: 83% 45 ± 10 NR	ACA, ATA, RNP	NR	ATA+ patients showed more severe nailfold changes (46% showed moderate capillary loss and 54% severe capillary loss) compared to ACA+ patients (39% having dilations and 21% giant capillaries, 34% moderate capillary loss and 6% severe capillary loss) and RNP+ patients.
Cutolo, 2004 [19]	SSc patients, classified with ACR 1980 criteria or LeRoy criteria	*n* = 241 F: 94.1% 57 ± 15 5 (1–13)	ATA, ACA	ANA: indirect immunofluorescence (IIF) on HEp-2 cells SSc-specific autoantibodies: ELISA	ATA+ more frequent in “Active” and “Late” patterns on NVC than in “Early”.
Koenig, 2008 [11]	RP at risk for developing SSc	*n* = 784 F: 82.75% age: 39.6 ± 13 RP duration (years): 3 (1–7)	ACA, anti-Th/To ATA ARA	ANA: indirect immunofluorescence (IIF) on HEp-2 cells SSc-specific autoantibodies: ELISA	ACA and anti-Th/To predicted enlarged capillaries (HR 6.64). ACA, anti-Th/To and ARA predicted capillary loss (HR 2.62) ACA predicted capillary teleangectasias (HR 3.1)
Sulli, 2013 [20]	SSc patients, classified with 1980 LeRoy criteria	*n* = 42 NR 47 ± 19 1 (IQR 3)	ACA, ATA	ANA: indirect immunofluorescence (IIF) on HEp-2 cells ACA and ATA: ELISA	ATA more often present in “late” pattern than in “early” and “active”. Non-significant associations between NVC patterns with other ANA patterns.
Ingegnoli, 2013 [30]	SSc patients classified according to the presence of clinical features, NVC patterns and autoantibodies	*n* = 2754 F: 87.15% age: 54.97 ± 13.6 disease duration: 7.62 ± 7.38	ACA, ATA	ANA: indirect immunofluorescence (IIF) on HEp-2 cells SSc-specific autoantibodies: ELISA	ATA more often present in the” late” pattern than in “early” and “active”
Pizzorni, 2017 [21]	SSc patients, classified with ACR 2013 criteria	*n* = 33 F: 84.8% 59 ± 21 years Mean SSc duration 6.6 ± 5 years)	ACA, ATA	ANA: indirect immunofluorescence (IIF) on HEp-2 cells SSc-specific autoantibodies: ELISA	“Early” and “active” pattern more often present in ACA patients, late pattern more often present in ATA patients.
Tieu, 2018 [22]	SSc patients classified according to the presence of clinical features, NVC patterns and autoantibodies	*n* = 152	ACA, ATA, RNP, RNAPIII	NR	ARA+ showed a higher grade of capillary damage compared with ACA and RNP+ (*p* < 0.001); ATA and ARA had a higher capillary dropout compared with ACA
Lambova, 2022 [23]	SSc patients, clinically diagnosed according to the extent of cutaneous involvement (limited vs. diffuse)	*n* = 19 51.56 ± 15.07	13 SSc-related autoantigens: ATA, CENP A, CENP B, RP11/RNAP-III, RP155/RNAP-III, fibrillarin, NOR-90, Th/To, PM-Scl100, PMScl75, Ku, PDGFR and Ro-52	ANA: indirect immunofluorescence (IIF) on HEp-2 cells Line immunoblot assay for detection of SSc-specific autoantibodies	ATA: associated with a lower mean capillary density and with a a higher frequency of “active” and “late” patterns. ARA: associated with a higher mean capillary density. No active and late patterns. Only one patient with early pattern.
Caramaschi, 2007 [24]	SSc patients, classified with ACR 1980 criteria	*n* = 103 F: 88.3% 54.3 ± 13.6 7 (1–46)	ACA, ATA	Anticentromere antibodies (ACA) were tested by indirect immunofluorescence on HEp-2 cells; anti-Scl70 antibodies were determined by ELISA	Non-significant associations between ACA, ATA and “early”, “active” and “late” scleroderma patterns.
Fichel, 2014 [26]	SSc patients, classified with 1980 LeRoy criteria	*n* = 88 NR 54.9 ±16.1 Duration of RP: 17.2 ± 14.8	ACA, ATA	ANA: indirect immunofluorescence (IIF) on HEp-2 cells SSc-specific autoantibodies: ELISA	Non-significant associations between autoantibodies and NVC patterns (defined as normal or SSc pattern).
Ghizzoni, 2015 [27]	SSc patients, classified with ACR 2013 criteria	*n* = 275 F: 90.1% 54.9 ± 14.2 Disease duration (months): 36.9 ± 65.5	ACA, ATA	ANA: indirect immunofluorescence (IIF) on HEp-2 cells SSc-specific autoantibodies: ELISA	Non-significant associations between autoantibodies and NVC patterns (defined as normal or SSc pattern).
De Santis, 2016 [25]	SSc patients, classified with ACR 2013 criteria	*n* = 44 F: 95.4% 66 (34–80) 9 (1–16)	ACA, ATA	ACA and ATA: ELISA	Non-significant associations between autoantibodies and “early”, “active” and “late” scleroderma patterns on NVC
Markusse, 2017 [28]	SSc patients, classified with ACR 2013 criteria or LeRoy criteria	*n* = 287 F: 82% 57 ± 14 3 (0.6–9)	ACA, ATA, RNAPIII, RNP, U3 RNP, Pm/Scl, Th/To, Ku	ANA: indirect immunofluorescence (IIF) on HEp-2 cells fluorescence ELISA	Non-significant associations between autoantibodies and “early”, “active” and “late” scleroderma patterns on NVC

Green background: significant associations, red background: non-significant association, SSc: systemic sclerosis, ACA: anti-centromere antibodies, ATA: anti-topoisomerase-I antibodies, RNP: anti-ribonucleoprotein, RP: Raynaud’s phenomenon, HR: Hazard ratio, NR: not reported, ACR: American College of Rheumatology, RNAP-III: anti-RNA polymerase III antibodies, ANA: antinuclear antibodies, HEp-2 cells: human laryngeal cancer cells, ELISA: enzyme linked immunosorbent assay, CENP-B: anti-centromeric B protein, NVC: nailfold videocapillaroscopy, NR: not reported, F: females, RP: Raynaud’s phenomenon.

**Table 2 antibodies-12-00003-t002:** Pre-clinical studies investigating the relationship between autoantibodies and microvascular damage/fibrosis.

Author, Year [Reference]	Sample Size (*n*) Classification Criteria Sex (F: %) Age, Average (Range) Duration of Disease or Symptoms in Years, Average (Range)	Autoantibodies	Immunoassays	Substrate	Results
Svegliati, 2017 [39]	*n*: 11 ACR/EULAR 2013 F: 81.9% Age: 56 (43–73) Disease duration: 7 (2–21)	Agonist Anti-PDGFRα	Fluorescence microscopy, FACS analysis	Human pulmonary artery smooth muscle cells	Anti-PDGFRα increased ROS production, NOX4 and mTORC1 expression in HPASMC inducing them to a synthetic state.
Raschi, 2020 [37]	*n*: 12 ACR/EULAR 2013 F: 100% Age: 47 (31–55) Disease duration: 30 (26–33)	ACA, ATA, ARA, anti-Th/To	IIF, chemiluminescent immunoassays, flow-cytometry analysis	Endothelial cells and fibroblasts	SSc-Ab-ICs induce a pro-inflammatory and pro-fibrotic endothelial cells phenotype. ACA-ICs, anti-Th/To-ICs: ↑ ICAM-1 All SSc-ICs but anti-Th/To-ICs augmented IL-8 levels ATA-ICs and anti-Th/To-ICs: ↑ ET-1 All SSc-ICs but ARA-ICs: ↑ TGF-Beta1 ATA-ICs and ACA-ICs: ↑ TLR-2, TLR-3 and TLR-4 on endothelial cells whereas anti-Th/To-ICs ↑ TLR9. Fibroblasts stimulated with with SSc-IC: ↑ TGF-beta1, alfa-SMA. Specifically, ATA-IC and ACA-IC: ↑colα1 and ACA-ICs: ↑ IL-6
Fonseca, 2006 [38]	*n*: 205 ACR 1980 criteria F, age and disease duration not reported	ACA, ATA, ARA	NR	NR	ARA+ patients showed a higher frequency of polymorphisms for EDNRA compared with ARA- and HCs (*p* < 0.05)
Corallo, 2019 [36]	*n*: 20 ACR 2013 criteria F: 80% Age: NR Disease duration: 10 (2–15)	ACA, ATA	IIF	Dermal fibroblasts	Pro-fibrotic activation in the human dermal fibroblasts through an hyper-expression of pro-fibrotic genes (increased mRNA of ACTA2, COL1A1 and TAGLN) and upregulated protein synthesis of α-SMA, Col-1 and SM22
Robitaille, 2009 [31]	ACR 1980 criteria Other data NR	ACA	ELISA	PASMCs (human pulmonary artery smooth muscle cells	Extracellular CENP-B autoantigens bind to human pulmonary artery smooth muscle cells (SMCs) stimulating their migration and their production of IL-6 and IL-8.
Robitaille, 2007 [32]	ACR 1980 criteria Other data NR	CENP-A and CENP-B	IIF, ELISA	Human pulmonary artery SMCs, normal human lung fibroblasts (NHLFs) and human pulmonary artery ECs	CENP-B does not bind fibroblasts or endothelial cells CENP-B released from apoptotic ECs was found to bind to SMCs.
Arcand, 2012 [34]	NR	ATA	ELISA	HDFs	The autoantigen topo I stimulated fibroblast migration via a G(αi) protein-coupled receptor, acting physiologically as a danger signal for the immune system to facilitate repair. CCR7 was found to interact directly with topo I.
Arcand 2012 [35]	NR	ATA	ELISA	HDFs	Topo I binds specifically to heparan sulfate proteoglycans on fibroblast surfaces and that anti-topo I autoantibodies from SSc patients amplify topo I binding to HS chains. The accumulation of topo I on cell surfaces by anti-topo I autoantibodies could contribute to the initiation of an inflammatory cascade stimulating the fibrosis.
Henault, 2004 [40]	99 SSc ACR 1980 criteria Age, sex and disease duration NR	AFA, ATA	ELISA + immunoblot	Fibroblast Endothelial cells Human pulmonary artery smooth muscle cells	AFAs in SSc are strongly correlated with ATA which themselves display AFA activity by reacting with determinants at the fibroblast surface.
Henault, 2006 [33]	37 SSc ACR 1980 criteria	ATA	IIF ELISA Immunobloting	Fibroblasts Endothelial cells Smooth muscle cells	The autoantigen topo I was found to bind specifically to fibroblasts in a dose-dependent manner, being recognized by ATA of SSc patients. The binding of ATA stimulated the adhesion and activation of cocultured monocytes. Topo I released from apoptotic endothelial cells was also found to bind specifically to fibroblasts.

PDGFRα: platelet derived growth factor alpha, FACS: Fluorescence activated cell sorting, HPASMC: human pulmonary artery smooth muscle cells, mTORC1: mammalian target of rapamycin complex 1, NOX4 (NADPH Oxidase 4), IIF (indirect immunofluorescence), SSc-Ab-ICs: SSc- related autoantibodies immune-complexed with autoantigens, ICAM-1: Intercellular Adhesion Molecule 1, TGFbeta: tumor growth factor beta, TLR: Toll-like receptor, ET-1: endothelin-1, ELISA: enzyme linked immunosorbent assay, CCR-: C-C Motif Chemokine Receptor, HDFs: human dermal fibroblasts, CENP-B: centromeric protein B. For the other abbreviations see the legenda of Table 1.

## Data Availability

No new data were created or analyzed in this study. Data sharing is not applicable to this article.

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
