# Peer review of "Specific Autoantibodies and Microvascular Damage Progression Assessed by Nailfold Videocapillaroscopy in Systemic Sclerosis: Are There Peculiar Associations? An Update"

_2073-4468, 2023, doi:10.3390/antib12010003_

Round 1

Reviewer 1 Report

The manuscript is very well written.

I have several recommendations to the authors:

1. "Mеthods" part in the abstract should be described with more details. 

2. In the introduction part the following paragraph should include examples for the "other" autoantibodies, or at least for the most important of them. 

"With the advent of more sensitive multiplexed immunoassays, other autoantibodies  have been described with a varying sensitivity and specificity in SSc cohorts depending on the ethnicity, geographic region, immunogenetic markers and autoantigen/immunoassay used[12,13].

3. The conclusion should be summarized and focus on two main points: 

1)Autoantibodies and NVC findings are serum and morphological biomarkers which stratify the prognosis of SSc patients. . Particularly, a faster microvascular progression and certain autoantibody profiles (i.e., ATA+ or ARA+ patients) characterize patients with  worse clinical outcomes.

2) The last 3 paragraphs is better to be summarized in one conclusion.

Author Response

We would like to thank reviewer 1 for appreciating our paper and for the constructive comments, which are addressed below.

  1. "Mеthods" part in the abstract should be described with more details

This section of the abstract is now described in more details. The following sentence has been added: A narrative review was conducted by searching relevant keywords related to this field in Pubmed, Medline and EULAR/ACR conference abstracts with a focus on the findings published in the last 5 years

  1. In the introduction part the following paragraph should include examples for the "other" autoantibodies, or at least for the most important of them. 

"With the advent of more sensitive multiplexed immunoassays, other autoantibodies  have been described with a varying sensitivity and specificity in SSc cohorts depending on the ethnicity, geographic region, immunogenetic markers and autoantigen/immunoassay used[12,13].

Thank you. The following paragraph has been added:

For instance, anti-Th/To define a phenotype of patients often characterized by limited cutaneous disease and a variable rate of cardio-pulmonary involvement (i.e. interstitial lung disease [ILD] or pericarditis) [12]. Many other autoantibodies could be considered markers of overlap syndrome with idiopathic inflammatory myopathies (IIMs). Anti-Ku are found in 2–7% of SSc patients presenting also clinical features of polymyositis (PM) or PM-Scl autoantibodies which define SSc patients with high frequency of ILD, calcinosis, dermatomyositis skin alterations and severe muscle involvement. Finally, novel SSc spe-cific autoantibodies have been described in about 10% of the former called “seronegative” SSc patients: anti-elF2B, anti-RuvBL1/2 complex, anti-U11/U12 RNP, and anti-BICD2 de-pict specific SSc subtypes with severe organ complications [12].

  1. The conclusion should be summarized and focus on two main points: 

1)Autoantibodies and NVC findings are serum and morphological biomarkers which stratify the prognosis of SSc patients. . Particularly, a faster microvascular progression and certain autoantibody profiles (i.e., ATA+ or ARA+ patients) characterize patients with  worse clinical outcomes.

2) The last 3 paragraphs is better to be summarized in one conclusion.

Thank you! The conclusive paragraph has been shortened and summarized as suggested.

Reviewer 2 Report

Specific autoantibodies and microvascular damage progression assessed by nailfold videocapillaroscopy in systemic sclerosis: are there peculiar associations? An update

This is a non-systematic review article that focus on the potential relationship between nailfold capillaroscopy changes in patients with systemic sclerosis and their association with antibodies, from a clinical and a basic science point of view.

This is a well written and very fluent manuscript. I congratulate the authors for this effort. This manuscript adds relevant information to the body of evidence of the usefulness of nailfold capillaroscopy in patients with systemic sclerosis as a non-invasive tool to establish an early diagnosis and possibly prognosis. One could infer that it may provide a surrogate of prognosis based on antibody positivity.

I suggest the following:

Abstract

Well written and clear

Introduction

Well written and clearly highlights the relevance of the study.

Materials and methods

Please include a short description of inclusion and exclusion criteria. Please explain how did you perform the article selection (paired selection? Only one author?, etc)

Results

Although I acknowledge that the search strategy was not intended to address specific information about the included antibodies, I kindly suggest to the authors to include a brief paragraph that describes what are the specific antibodies related to systemic sclerosis, how are they measured and interpreted. Including this information would allow a larger share of readers to benefit from this manuscript. In addition, to highlight the differences in laboratory methods to measure their positivity would allow the reader to gain insight in the heterogeneity of this field.

As the authors are internationally-renowned experts in nailfold capillaroscopy, I would suggest to include a brief paragraph on this technique and its results; I would encourage the inclusion of a figure with nailfold abnormalities. This ingredient will allow a wider context of this technique and will allow the reader to associate the excellent literature review findings with actual abnormalities in the “physical exam”.

Line 94-97: this sentence affirms that the ATA+ patients may present a faster progression of microangiopathy. Nonetheless, the arguments to support this statement should be more robust. Please elaborate on this interesting phenomenon.

Line 98-104: on the other hand, this paragraph suggests specific phenomenon associated with early/active vs late subgroups, whereas line 94-97 found a similar result but in the “subgroup” of early vs active/late. I would suggest to elaborate on these differences as I would consider that neither “early” is a synonym of “active” changes nor “active” a synonym of “late” changes.

Anti-Th/To antibodies are mentioned but their relevance for systemic sclerosis are not explained, please elaborate on this issue. This could be explained in the brief paragraph on autoantibodies I suggested earlier.

Line 155-164: the authors comment that:

“Extracellular centromeric protein B (CENP-B) autoantigens were found 156 to specifically bind the surface of human pulmonary artery smooth muscle cells 157 (PASMCs) stimulating their migration and their production of IL-6 and IL-8, displaying 158 an important physiological role in the local tissue repair”

In the next paragraph, affirm:

“Since CENP-B behaves as 159 a potent stimulation of PASMCs activating the endothelial growth factor receptor (EGFR), 160 anti–CENP-B antibodies were found to prevent CENP-B from transactivating EGFR and 161 exerting its cytokine-like activities toward vascular smooth muscle cells”

However, based on the known effects of IL-6 (pro-inflammatory properties, such as Th17 differentiation, cytotoxic T-cell differentiation, antibody production, blunt Treg differentiation and even promote fibrosis) and IL-8 (neutrophil chemotactic properties, that ultimately promote inflammation), I find a little counterintuitive that the fact of “blocking” their production that “(...) shed light on the possible role of CENP-B/ACA immune complexes in 163 the aberrant tissue repair processes observed in SSc patients”

Please elaborate on this phenomenon. I can’t rule out that I’m not having a correct understanding of this paragraph. If that is the case, I suggest to reorganize the idea to make it more understandable.

Table 1: “ARA” definition is missing in the table footnote

Figure 1: please adjust in light of the comment made on the paragraph between line 155-164

Discussion and conclusion

Well written and clear

Author Response

Many thanks to reviewer 2 for for appreciating our work and for the precious and detailed feedback which is addressed as follows:

Please include a short description of inclusion and exclusion criteria. Please explain how did you perform the article selection (paired selection? Only one author? etc)

In the methods section, a further paragraph has been added explaining in details the research process (line 96-102).

I kindly suggest to the authors to include a brief paragraph that describes what are the specific antibodies related to systemic sclerosis, how are they measured and interpreted. Including this information would allow a larger share of readers to benefit from this manuscript. In addition, to highlight the differences in laboratory methods to measure their positivity would allow the reader to gain insight in the heterogeneity of this field.

From line 69 to 87 these concepts have been integrated in the manuscript. Thank you!

As the authors are internationally-renowned experts in nailfold capillaroscopy, I would suggest to include a brief paragraph on this technique and its results; I would encourage the inclusion of a figure with nailfold abnormalities. This ingredient will allow a wider context of this technique and will allow the reader to associate the excellent literature review findings with actual abnormalities in the “physical exam”.

This is done from line 60 to 65 and figure 1 on NVC findings in SSc was added.

Line 94-97: this sentence affirms that the ATA+ patients may present a faster progression of microangiopathy. Nonetheless, the arguments to support this statement should be more robust. Please elaborate on this interesting phenomenon.

The conclusion of the authors is now reinforced. Of course, the arguments need to be more robust to generalize this concept but, in this sentence, we just reported the study of Cutolo et al (Rheumatology 2004) trying to interpret a bit more, as you suggested, these findings but only for the single study.

Line 98-104: on the other hand, this paragraph suggests specific phenomenon associated with early/active vs late subgroups, whereas line 94-97 found a similar result but in the “subgroup” of early vs active/late. I would suggest to elaborate on these differences as I would consider that neither “early” is a synonym of “active” changes nor “active” a synonym of “late” changes.

Thank you, this paragraph has been clarified and described in more details from line 140 to line 142.

Anti-Th/To antibodies are mentioned but their relevance for systemic sclerosis are not explained, please elaborate on this issue. This could be explained in the brief paragraph on autoantibodies I suggested earlier.

Since this was requested also by reviewer 1, a short paragraph on SSc-specific autoantibodies has been created from line 69 to 78 including Th/To autoantibodies.

Line 155-164: the authors comment that:

“Extracellular centromeric protein B (CENP-B) autoantigens were found 156 to specifically bind the surface of human pulmonary artery smooth muscle cells 157 (PASMCs) stimulating their migration and their production of IL-6 and IL-8, displaying 158 an important physiological role in the local tissue repair”

In the next paragraph, affirm:

“Since CENP-B behaves as 159 a potent stimulation of PASMCs activating the endothelial growth factor receptor (EGFR), 160 anti–CENP-B antibodies were found to prevent CENP-B from transactivating EGFR and 161 exerting its cytokine-like activities toward vascular smooth muscle cells”

However, based on the known effects of IL-6 (pro-inflammatory properties, such as Th17 differentiation, cytotoxic T-cell differentiation, antibody production, blunt Treg differentiation and even promote fibrosis) and IL-8 (neutrophil chemotactic properties, that ultimately promote inflammation), I find a little counterintuitive that the fact of “blocking” their production that “(...) shed light on the possible role of CENP-B/ACA immune complexes in 163 the aberrant tissue repair processes observed in SSc patients”

Please elaborate on this phenomenon. I can’t rule out that I’m not having a correct understanding of this paragraph. If that is the case, I suggest to reorganize the idea to make it more understandable.

Thanks a lot for your precise comment! Despite being aware that the aberrant release of IL-6 and IL-8 generates an abnormal immune response with the phenomena that you described, we meant that the local pro-inflammatory environment physiologically promotes in the first stages the healing of wounds, which is in this case the first step of the endothelial tissue repair.

The authors of that translational study concluded that, normally, autoantigenic CENP-B release activates an EGF-R mediated pathway with the release of local IL-6 and IL-8 which promote in part the endothelial repair in the first stages of tissue repair (the first phase of wound healing is “inflammatory” with the recruitment of innate immune cells and later there is deposition of collagen and extracellular matrix which form the scarring tissue).

Therefore, one of the potential mechanisms of ACA-related endothelial damage is the interference with this initial process of endothelial repair.

We fully agree with you that in the later steps of SSc pathogenesis, these cytokines are purely pro-inflammatory and promote autoimmunity and fibrosis (i.e. IL-6 whose inhibition by TCZ has shown interesting data in SSc-related ILD).

We hope that now in the paragraph 203-213 this is clearer. In Figure 2 this concept has been integrated as well!

Reviewer 3 Report

This is great article summarizing qualitatively the results from the ssc-specific antibodies.and structural microvascular abnormalities evaluated 72 with NVC, discussing the evidence in light of the most recent literature 

In the figure anti-ATIR antibodies production triggered by produced by IL-8 and IL-6 ?.

Author Response

We would like to thank reviewer 3 for the nice comments.

  1. In the figure anti-ATIR antibodies production triggered by produced by IL-8 and IL-6 ?.

If the concept was misleading, the figure has now been edited! Physiologically, it appears that centromeric protein B released by apoptotic endothelial cells promotes the release of IL-6 and IL-8 which have a role in tissue repair. ACA might block this process leading to endothelial damage and disruption of endothelial repair process.

We hope this concept is clearer in the new edited figure. Thanks again!